# GQA: GENERATION QUALITY ASSESSMENT OF AIGC VIDEOS BASED ON HUMAN ASSESSMENT: DATASET, SCORING AND EXPLANATION

## ABSTRACT

Recent advances have significantly elevated the quality of AI-generated videos; however, existing evaluation metrics still struggle to align closely with human perceptual judgments. While prior work has repurposed deep learning models or borrowed algorithms from other domains to assess generative content, their outputs often exhibit noticeable discrepancies with real human evaluations. To address this critical gap, we introduce the GQA dataset — a human-aligned benchmark comprising: (1) videos generated by dozens of state-of-the-art models, including those from the VAE and Diffusion Model (DM) families; (2) dozens of refined evaluation metrics systematically categorized into three core dimensions — Video-Text Consistency, Realism, and Traditional Quality; and (3) a prompt-adaptive metric selection mechanism that ensures evaluations are contextually relevant, avoiding misaligned assessments across semantically unrelated dimensions. GQA enables more accurate, interpretable, and perception-aware evaluation of AI-generated video content.

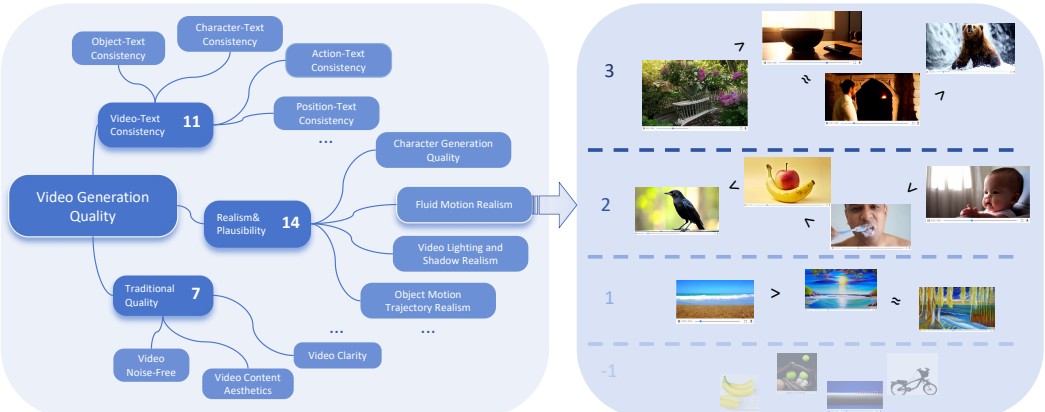

Figure 1: Assesstion Dimensions (Left) & Construction Methodology (Right). The GQA dataset encompasses 32 evaluation metrics across three major categories: Consistency, Realism, and Traditional Quality. The data construction for each metric follows a structured methodology: Initial Categorization: Videos are first categorized based on relevant attributes. Intra-class Pairwise Comparison: Videos within each category are then comparatively evaluated. Quality-ranked Scoring: This process ultimately yields quality-ranked scores for the generated videos.

## 1 INTRODUCTION

The rapid advancement of AI-generated video technology (e.g., diffusion models like Sora Zheng et al. (2024) and autoregressive frameworks like VideoPoet Kondratyuk et al. (2023)) has revolutionized content creation. However, evaluating the quality of these synthetic videos presents unique

challenges fundamentally distinct from traditional Video Quality Assessment (VQA). Existing approaches can be categorized into three paradigms:

**Traditional Video Quality Assessment (VQA).** Classical VQA methods focus on quantifying distortions introduced during compression or transmission. Full-reference metrics like PSNR and SSIM measure pixel-level fidelity but fail to capture semantic coherence, which is crucial for AI-generated content. No-reference methods like NIQE Mittal et al. (2013) leverage natural scene statistics, yet their reliance on handcrafted features limits applicability to synthetic artifacts (e.g., inconsistent object scaling in generated videos ). Hybrid models like VMAF integrate machine learning to predict human perception but remain constrained by their design for natural video degradation patterns.

**Video Aesthetic Quality Assessment (VAQA).** Aesthetic assessment extends beyond technical fidelity to evaluate artistic merit, encompassing composition, color harmony, and emotional impact. Datasets like AVA Murray et al. (2012) provide human-annotated aesthetic scores, while deep learning frameworks integrate low-level features (e.g., saliency maps) with high-level semantics. For AI-generated videos, it remains fundamentally challenging to establish whether machines can learn subjective aesthetics and how to balance creative expression with normative standards (e.g., surrealism versus photorealism).

**Video Generation Quality Assessment (VGQA).** Emerging metrics specifically target the unique flaws of synthetic videos: **Temporal Consistency**: Fréchet Video Distance (FVD) Unterthiner et al. (2018) compares feature distributions between real and generated videos using pre-trained 3D CNNs, penalizing unnatural motion. **Semantic Accuracy**: CLIPScore Hessel et al. (2021) aligns video content with text prompts via contrastive language-vision models, addressing hallucinations (e.g., "a dog flying in space" violates physical laws). **Artifact Detection**: Recent work Ojha et al. (2023) trains detectors on GANs producing facial distortions or diffusion model-specific noise residuals. Despite progress, current metrics still struggle with long-term coherence (e.g., graphic continuity in minute-long videos) and multimodal alignment (e.g., audio-visual synchronization). Current AI video generation efforts predominantly focus on enhancing realism. Recent years have witnessed explosive growth in novel video generation models emerging from diverse academic institutions and enterprises. Concurrently, significant work advances the evaluation of these generative models. Notable mature frameworks include VBench Huang et al. (2024a)—a comprehensive video generation assessment system—and Gaia Chen et al. (2024b), an evaluation benchmark specifically targeting motion.

Our objective is to pioneer a distinct direction: advancing video generation assessment through human-perception-driven methodologies. So we integrate objective video quality, subjective aesthetic quality, and AI-generated quality assessment to propose the GQA dataset. The key features of GQA are:

- Construction of 32 targeted evaluation dimensions categorized into three major classes: video-text consistency, realism and plausibility of video content, and traditional video quality.

- Text-content correlated evaluation where assessment dimensions are assigned based on textual descriptions and actual video content.

- Human-centered annotation scheme combining coarse-grained classification and fine-grained comparative labeling to deliver efficient yet accurate dimensional evaluations.

## 2 RETHINKING VIDEO GENERATION QUALITY ASSESSMENT

With the rapid development of AI-generated video technology, evaluating the quality of generated content has become a key research focus. Existing evaluation methods primarily rely on a combination of automated metrics and human subjective ratings, among which comprehensive evaluation frameworks like VBench Huang et al. (2024a) have garnered significant attention in both academia and industry. This section systematically reviews the VBench-centered evaluation system and compares it with other representative works.

VBench, an open-source evaluation framework jointly proposed by Peking University and Microsoft Research, aims to address the challenge of multi-dimensional quality assessment for generated

Table 1: Number and explanation of different assessment dimensions

| Type | Num. | Assessment Dimension | Description |
|---|---|---|---|
| Video-Text Consistency | 4 | Character-Text Consistency | Whether specific characters in the video match the text description (e.g., Elon Musk should appear as the correct individual). |
| | 1 | Action-Text Consistency | Whether actions in the video match the text description (e.g., running, jumping), focusing solely on the action regardless of the subject. |
| | 17 | Scene-Text Consistency | Whether scenes in the video match the described settings (e.g., hospital, school), including identifiable scene elements. |
| | 14 | Object Position-Text Consistency | Object positions refer to relative placement based on camera orientation (e.g., if "a motorcycle is to the left of a bus," they should appear on corresponding sides of the video frame). |
| | 12 | Object Attribute-Text Consistency | Object attributes include descriptive features like color, shape, and texture. |
| | 11 | Object-Text Consistency | Whether objects in the video can be correctly identified as those mentioned in the text. |
| | 25 | Video Content-Text Consistency | Overall alignment where every textual description should be accurately generated. |
| | 29 | Video Speed-Text Consistency | Whether video speed matches textual descriptions (current samples only include slow-motion). |
| | 30 | Video Style-Text Consistency | Whether artistic styles mentioned in text (e.g., Van Gogh, Picasso) are recognizable in the video. |
| | 3 | Camera Movement-Text Consistency | Whether camera movements described in text (e.g., pan left, tilt right) are properly executed. |
| | 23 | Unrealistic Description Imaginative Presentation | When text describes unrealistic scenarios (e.g., "an astronaut riding a horse in space"), whether the video presentation aligns with imaginative expectations. |
| Realism & Plausibility | 16 | Rigid Body Collision Realism | Whether rigid body collisions in videos appear physically plausible. |
| | 2 | Action Realism | Whether actions could realistically be performed. |
| | 18 | Scene Realism | Whether scenes appear sufficiently realistic when no special style is specified in text. |
| | 31 | Weather Representation Realism | Whether weather conditions appear realistic. |
| | 22 | Time Period Representation Realism | Whether time-period representations appear authentic. |
| | 8 | Gaseous Motion Realism | Whether gas dynamics (smoke, vapor) appear physically accurate. |
| | 7 | Fluid Motion Realism | Whether fluid movements appear physically plausible. |
| | 9 | Gradual Change Motion Realism | Whether gradual transformations (balloon inflation, plant growth) appear physically accurate. |
| | 13 | Object Motion Trajectory Realism | Whether object movement paths follow physically plausible dynamics. |
| | 15 | Object Realism | Whether objects appear sufficiently realistic. |
| | 5 | Character Generation Quality | Whether human characters appear sufficiently realistic. |
| | 21 | Textual Attribute Representation Realism | Whether object attributes (color, shape, texture) match real-world appearances. |
| | 27 | Video Lighting and SGQAow Realism | Whether lighting and sGQAows appear physically accurate. |
| | 10 | Moving Scene Reasonableness | Whether scene transitions during camera movements maintain proper perspective. |
| Traditional Quality | 6 | Entity Motion Naturalness | Objects should maintain natural form without distortion during movement. |
| | 26 | Video Content Aesthetics | Overall visual appeal of video content. |
| | 0 | Abnormal Lighting Detection | Videos should avoid lighting artifacts (overexposure, abnormal flares). |
| | 28 | Video Noise-Free | Videos should exhibit no noticeable noise artifacts. |
| | 24 | Video Clarity | Whether video resolution is sufficiently sharp. |
| | 19 | Static Content Non-distortion | Stationary objects shouldn't distort abnormally during camera movement. |
| | 20 | Static Content Stability | Stationary objects shouldn't distort abnormally over time (temporal consistency). |

videos. Its core design philosophy employs a hierarchically decoupled evaluation structure covering two primary dimensions: video quality and content consistency.

VBench 1.0 defined 16 evaluation metrics categorized into two classes: Video Quality: Including temporal consistency (Temporal Flickering), motion smoothness (Motion Smoothness), aesthetic quality (Aesthetic Quality), etc. Video-Condition Consistency: Encompassing text-video semantic alignment (Text-Video Alignment), spatial relationships (Spatial Relationship), object attributes (Object Attribute), etc. The evaluation system employs algorithms and models from diverse domains for different metrics, while correlating with human-annotated MOS (Mean Opinion Score) scores to enhance assessment reliability. However, real-world cases still reveal discrepancies between model evaluation results and human perception.

VBench++ Huang et al. (2024b) expanded the framework by introducing realism-centric metrics, including Physical Plausibility and Commonsense Consistency, resulting in 18 total metrics. For example: Physical Plausibility: Evaluates adherence to real-world physical laws (e.g., gravity, collisions, fluid dynamics), such as "whether a flying dragon's wing flap generates plausible air-flow disturbance." Commonsense Consistency: Detects logical flaws in generated content, such as "whether food realistically enters the mouth during a person's eating action." More sophisticated models—such as human pose detectors and physics simulation engines—were integrated into the evaluation pipeline, leading to significant improvements in accuracy.

Diverging from VBench's approach, we prioritize human-centric evaluation. We contend that the most direct and valuable method is to present AI-generated videos to humans for judgment. Our objective is to design protocols enabling humans to make effortless and unbiased assessments. Our methodology involves decomposing video evaluation into multiple dimensions, similar to VBench, aiming to provide clear guidance on specific aspects evaluators should focus on, thereby simplifying the assessment process. Recognizing the difficulty humans face with absolute scoring tasks, we leverage their comparative strength: direct choice between options. Consequently, our next step implements a pairwise comparison paradigm. Evaluators simply select the preferred video for a specific attribute from two presented options. This approach significantly lowers cognitive load, increases decision ease, and improves assessment accuracy.

## 3 DATASET DESCRIPTION

### 3.1 ASSESSMENT DIMENSIONS

As shown in the Table 1, GQA comprises 32 evaluation metrics spanning three major categories: consistency, realism, and traditional quality. For identical video content, we perform fine-grained segmentation according to different evaluation objectives. Taking the dual aspects of consistency and realism as an example: consistency ultimately requires the coupling between human cognition and textual descriptions, similar to the alignment of image and text modalities in CLIP, both being semantically grounded cognition. Realism concerns only the video itself without any relation to text. A simple example: for a video generating an apple based on text, the highest objective for evaluators under the consistency dimension is to effortlessly recognize the object in the video as an apple. Realism requires evaluators to perceive this apple as if it were a real apple seen in reality captured by a camera.

### 3.2 GENERATION MODEL

Our evaluation framework encompasses 35 generative models, including: Show-1 Zhang et al. (2023), StepVideo Ma et al. (2025a), CogVideo Yang et al. (2024), Vchitect Fan et al. (2025), Gen-2 Runway (2023), HiGen Qing et al. (2024), EasyAnimate Xu et al. (2024), AnimateDiff Guo et al. (2024), InstructVideo Yuan et al. (2024), VideoCrafter Chen et al. (2023; 2024a), RepVideo Si et al. (2025), IPOC Yang et al. (2025), Gen-3 Runway (2024), OpenSora Zheng et al. (2024), Lavie Wang et al. (2024c), Mochi Team (2024), HunyuanVideo Kong et al. (2024), Mira Ju et al. (2024), TFt2v Wang et al. (2024b), Pika Labs (2023), LTX-Video HaCohen et al. (2024), STIV Lin et al. (2024), Latte Ma et al. (2025b), AnimateLCM Wang et al. (2024a), MiniMax Minimax (2024), Wanx Wan et al. (2025), Luma AI (2024), Kling DeepSeek (2024). We obtain generated videos of these models from VBench's publicly shared Test Data. These models predominantly represent work from the

past two years while also including relatively early generative models. The publication dates of the models may serve as indicators of video generation quality to some extent.

Figure 2: Heatmap representing the comparative performance of various generative models across different dimensions. Each cell represents the normalized average score (in percentage) of a model for a specific dimension, with scores adjusted to reflect the proportion of the highest score within each dimension.

As illustrated in Figure 2, we computed the average scores of each generative model across all dimensions, normalized them into percentage scores based on the respective upper and lower bounds of each dimension, and visualized the results in a heatmap. The heatmap reveals that no single model excels across all dimensions; however, overall, newly released models outperform their older counterparts, and closed-source models generally surpass open-source ones.

## 3.3 ANNOTATION SCHEME

As shown in Figure 1, we employ a combined annotation methodology of classification followed by pairwise comparison.

In stage 1 we collect and filter prompt texts based on our designed assessment dimensions. Our annotation data is sourced from VBench's evaluation videos. Consequently, we classify the 945 prompts used by VBench. Each prompt may correspond to multiple assessment dimensions. For example, the prompt 'A beautiful coastal beach in spring, waves lapping on sand, zoom in' relates to dimensions like scenery, fluid motion, and camera movement. Through this classification, we gather varying numbers of videos and their corresponding prompts for each assessment dimension.

For each assessment dimension, we design specific questions with unique response options. These options represent quality levels (effectively serving as scores) within that dimension. Example (Object-Text Alignment): This dimension includes 4 options:

- **-1** (Invalid Question): A universal option present in most dimensions. Annotators select this to discard a sample when: The prompt for a "consistency" dimension lacks a specified target (e.g., object, scene). The prompt for a "realism" dimension intentionally describes an unrealistic scenario.
- **1** (Completely Inconsistent): The object exhibits no alignment with the text description.

- **2** (Partially Consistent): The object exhibits characteristics of the described target.
- **3** (Fully Consistent): The object perfectly matches the text description. This coarse-grained classification approach significantly reduces annotator burden, enabling them to perform accurate labeling efficiently.

In stage 2 we aggregate the Stage 1 annotation results and partition the data into groups sharing the same assessment dimension, same prompt text, and same initial classification label. Within each group, we generate all possible unique pairwise combinations of videos (from start to end). Annotators are presented with video pairs and simply judge which video is superior for the specific dimension in question, or if they are equivalent. Based on the pairwise comparison results, videos are ranked ordinally within their group for the dimension. Fine-grained scores are assigned within the range defined by the initial coarse class. For instance, videos initially classified as 1 (Completely Inconsistent) in Stage 1 are assigned specific scores within the interval [1, 2] based on their relative position in the Stage 2 ranking.

## 3.4 ANNOTATION TEAM

Prior to annotation, the team leader conducted a structured training session for 18 professional annotators from a certified annotation company, using theoretical instruction and case demonstrations to ensure a shared understanding of the annotation guidelines and promote consistency and accuracy.

A quality review panel—comprising the team leader and three senior annotators—was established to oversee annotation quality. The panel performed random audits of 25% of each batch; non-compliant batches were returned for revision. Common issues were addressed through representative examples, and annotators received real-time support, continuously improving both efficiency and reliability.

All tasks were carried out on an in-house annotation platform designed to isolate individual assignments, preventing cross-influence among annotators and ensuring the objectivity and independence of each annotation judgment.

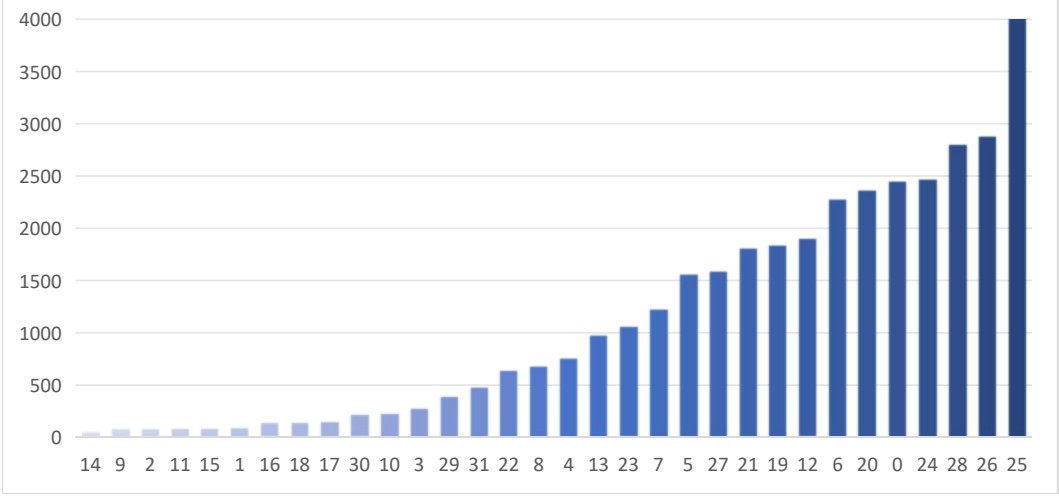

Figure 3: The data volume of different assessment dimensions in the dataset, among which the attribute of overall consistency has the highest value, which is 7669.

## 3.5 DATA DISTRIBUTION

As shown in Table 2, after the two-stage annotation process, the GQA dataset contains 9,474 videos from 945 prompts, resulting in a total of 39,724 merged human annotations.

The sample distribution across assessment dimensions in our dataset exhibits significant variation, ranging from 7,669 (highest) to 100 (lowest) samples per dimension as illustrated in the accom-

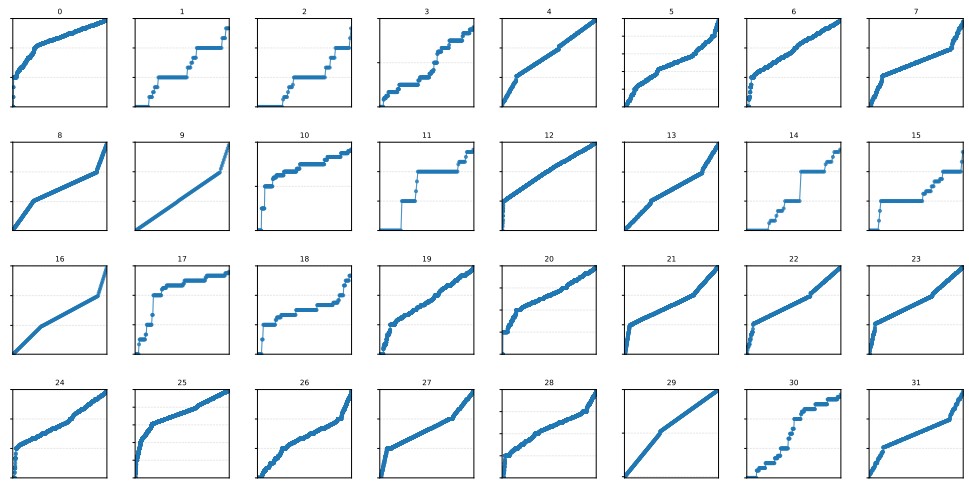

Figure 4: Distribution of Data Across Assessment Dimensions in the GQA Dataset. Serial numbers correspond to Table 1. The grid spacing in sub-tables represents a 1-point interval, with the minimum score being 1 point for each dimension.

Table 2: Statistics of the GQA dataset.

| Item | Count | Item | Count |
|---|---|---|---|
| Total Videos | 9474 | Total Prompts | 945 |
| Classification Annotations | 42594 | Pairwise Comparison Annotations | 31468 |
| Final Combined Annotations | 39724 | | |

panying Figure 3. Dimensions pertaining to Foundational Quality maintain a relatively substantial sample size overall, owing to their minimal dependence on specific video content. For Consistency dimensions directly corresponding to those in VBench, the availability of relevant prompts allowed sufficient coverage; we employed random sampling of prompts during annotation to prevent substantial disparities in sample sizes compared to other dimensions. Realism dimensions necessitate meticulous content scrutiny, revealing notable challenges: Rigid-body collisions prove exceedingly rare in the collected videos. Within the limited collision-related content available, elastic collisions predominate. However, due to the frequent absence of close-up shots explicitly depicting collision dynamics in these instances, evaluation focus primarily shifts to analyzing the motion trajectories of involved objects.

The scarcity of samples also leads to highly concentrated distributions, as illustrated in the accompanying Figure 4. Specifically, the dimensions of Time Period Representation Realism (22), Unrealistic Description Imaginative Presentation (23), Video Clarity (24), Video Speed-Text Consistency (29), and Weather Representation Realism (31) exhibit particularly clustered distributions.

## 4 BASELINE

As illustrated in Figure 5, we utilize VideoMAE Tong et al. (2022) for video encoding and CLIP Radford et al. (2021) for text encoding to construct a video generation quality assessment model, training dedicated baseline models for each evaluation dimension. During training, parameters of the VideoMAE and CLIP encoders remain frozen. The 3D features output by VideoMAE undergo channel fusion through a linear layer, then concatenate with CLIP's textual features for joint regression and classification training. Each baseline model is trained for 5 epochs on a single A100 GPU using the Adam optimizer with a learning rate of 1e-3 and a batch size of 64. Data for each evaluation dimension is split into 9:1 training-test sets.

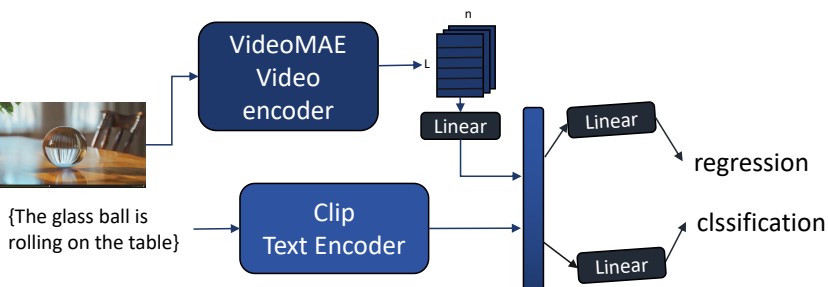

Figure 5: Baseline model structure.

Table 3: Baseline models test results. Serial numbers correspond to Table 1.

| Num. | MSE↓ | Acc.↑ % | Num. | MSE↓ | Acc.↑ % |
|---|---|---|---|---|---|
| 0 | 0.165 | 77.23 | 16 | 0.677 | 26.66 |
| 1 | 0.766 | 45.45 | 17 | 0.402 | 87.50 |
| 2 | 0.539 | 20.00 | 18 | 0.460 | 87.50 |
| 3 | 0.251 | 62.06 | 19 | 0.519 | 46.23 |
| 4 | 0.754 | 54.54 | 20 | 0.481 | 43.69 |
| 5 | 0.966 | 42.67 | 21 | 0.315 | 65.02 |
| 6 | 0.327 | 63.75 | 22 | 0.415 | 63.07 |
| 7 | 0.330 | 67.74 | 23 | 0.370 | 63.55 |
| 8 | 0.279 | 64.28 | 24 | 0.353 | 55.24 |
| 9 | 0.539 | 40.00 | 25 | 0.790 | 52.80 |
| 10 | 0.567 | 91.66 | 26 | 0.359 | 65.05 |
| 11 | 0.643 | 60.00 | 27 | 0.401 | 53.75 |
| 12 | 0.312 | 62.30 | 28 | 0.592 | 49.11 |
| 13 | 0.504 | 51.51 | 29 | 0.222 | 65.00 |
| 14 | 5.241 | 28.57 | 30 | 1.052 | 56.52 |
| 15 | 0.147 | 90.00 | 31 | 0.359 | 84.00 |

As shown in Table 3, we provide a minimal, proof-of-concept baseline to demonstrate the feasibility of learning from the GQA annotations. Specifically, we train a unified multimodal classification-regression multi-task model across all 32 evaluation dimensions, using the same architecture and hyperparameters for every task—without any dimension-specific tuning or architectural customization.

Unsurprisingly, performance varies widely: the best-performing dimensions achieve up to 90% accuracy and 0.14 MSE, while the worst drop to 20% accuracy. We attribute this gap to two key factors: (1) the heterogeneous nature of the evaluation dimensions—some require fine-grained temporal reasoning (e.g., fluid motion realism), while others rely on coarse semantic alignment (e.g., object-text consistency); and (2) significant imbalances in annotation volume across dimensions, which affect model convergence.

Importantly, this baseline is not intended to be competitive or optimal. Rather, it serves as a starting point to illustrate how GQA can be used for supervised learning. We anticipate that future work will develop specialized architectures, incorporate modality-specific pretraining, or leverage human feedback loops to achieve robust performance across all dimensions. Our goal here is not to solve the assessment problem, but to enable and encourage such research through a human-aligned, fine-grained benchmark.

## 5 CONCLUSIONS

This paper proposes a novel evaluation framework for AI-generated video quality assessment, establishing the human perception-driven multidimensional video generation quality assessment dataset GQA. Guided by three primary categories—video-text consistency, realism & plausibility of video

content, and fundamental video quality—GQA encompasses 12 consistency-related dimensions, 13 Realism-Plausibility dimensions, and 7 fundamental quality dimensions, comprehensively covering key aspects of AI-generated videos. We further design an annotation-efficient evaluation protocol combining categorical and comparative labeling, significantly enhancing annotation efficiency and accuracy. Finally, we construct and train baseline models based on video-text encoders, demonstrating the dataset's feasibility for model training. Future work will address data imbalance and insufficiency across dimensions in GQA while expanding evaluation coverage, ultimately establishing a large-scale human-centric AI video assessment system.

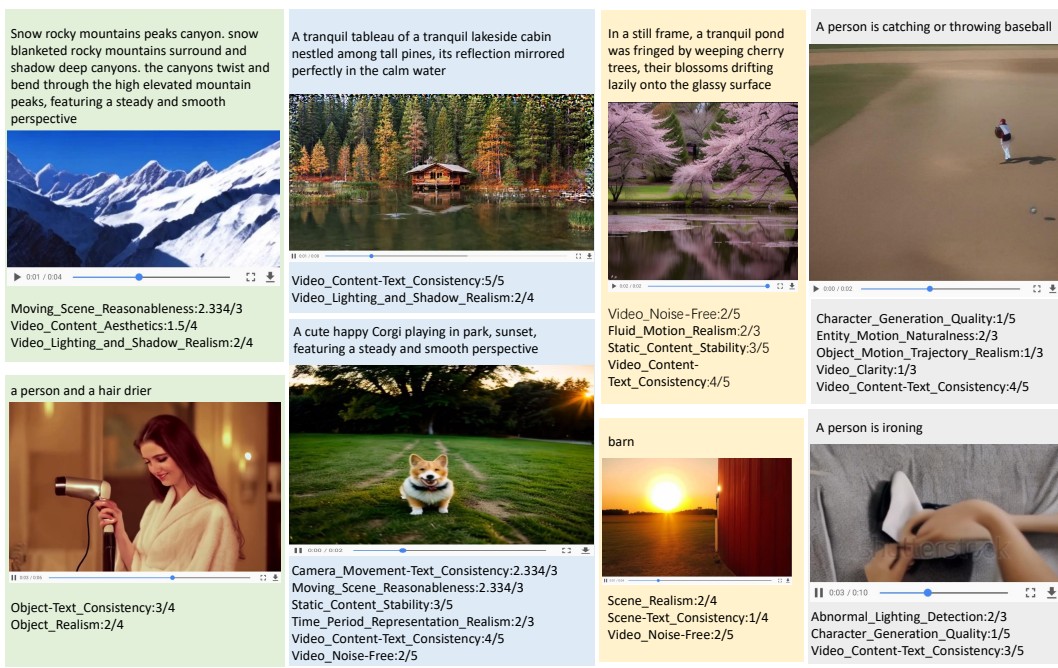

Figure 6: Data instances in the GQA Dataset, where a single video may correspond to multiple assessment dimensions.

ETHICS STATEMENT

This study adheres strictly to ethical research standards and has received retrospective approval from the ethics committee of an affiliated institution. All annotators were professional annotators recruited from a certified annotation company, possessing relevant expertise and participating voluntarily. Prior to annotation, an online informed consent process clearly communicated the study's purpose, task procedures, potential risks (e.g., time commitment and fatigue due to repetitive tasks), and the right to withdraw at any time without penalty. All annotated materials consisted of de-identified public video clips containing no personally identifiable information, private data, or sensitive content (e.g., violence or explicit material). The annotation platform recorded only anonymous annotator IDs, with no linkage to real identities. To uphold labor rights, annotators could freely schedule their work hours, faced no compulsory participation, and received fair compensation for any required revisions based on actual effort. A quality assurance mechanism and technical support team were also in place to promptly address annotation issues and platform malfunctions, ensuring a fair, safe, and transparent process.

In this paper, Large Language Models (LLMs) were used as an assistive tool during the writing process, primarily to aid in language polishing and improve clarity and fluency. All core ideas, research design, data analysis, and conclusions were independently developed by the authors. The LLM was not involved in any academic judgment or substantive content generation. Further details on its use are provided in the main paper.

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
