# OpenReview forum: "GQA: Generation Quality Assessment of AIGC Videos based on Human Assessment: Dataset, Scoring and Explanation"
_ICLR.cc/2026/Conference — ICLR 2026 Conference Withdrawn Submission_

### Official Review · Reviewer_niTX · 2025-10-19

**Soundness:** 2
**Presentation:** 1
**Contribution:** 2
**Rating:** 2
**Confidence:** 5

**Summary:**

This paper introduces a human-aligned dataset for AI-generated videos, GQA, which comprises 32 evaluation metrics with 9,474 videos from 945 prompts, together with 39,724 merged human annotations. An annotation-efficient evaluation protocol combining categorical and comparative labeling is proposed to enhance annotation efficiency and accuracy.

**Strengths:**

The GQA encompasses 35 generative models and 32 evaluation metrics, which is comprehensive.

**Weaknesses:**

a). The writing and organization of this paper is poor, making it hard to read:

    1. The design of Figure 1 is indeed somewhat confusing. After all, it is an overall overview of this project. It is difficult to understand the “Construction Methodology” (Right) through such simple presentation. What kind of comparison does the symbol in the picture represent? Why use this particular 3, 2, 1, -1 grading system?
    2. The introduction section devoted a significant amount of space to reviewing existing work, but failed to provide a detailed explanation of the motivation behind this study and a clear description of the work itself.
    3. Table 1 should be moved to appendix. Is column “number” means index?
    4. I suggest that in Section 3.2, the introduction of generative models be presented in chronological order for better readability.
    5. The information density in Figure 2 is too high. It is recommended to describe each aspect separately according to the three types of evaluation proposed.
    6. Why adopt labels such as -1, 1, 2, 3? Is there any theoretical basis for this? Why not use 0, 1, 2, 3 or other commonly used grading systems?
    7. What do the x- and y- axes in Figure 4 represent?

b). The contribution is limited, lacking original technical contributions:

    1. Section 2, although titled "Rethinking", merely provides a descriptive review of the vbench series, lacking in-depth perspectives. The proposed human-centered evaluation has already been presented in existing work. The AI-generated video evaluation based on subjective assessment is a mainstream approach.
    2. The data volumn in Fig. 3 exhibits significant long-tailed distribution, which may introduce bias into the evaluation results.
    3. The technical contribution of the proposed baseline is poor.

c). Lack of experimental analysis:

    1. Lack details of human annotations. See Details Of Ethics Concerns.
    2. Given that this paper has drawn upon the design of vbench, it is necessary to conduct a comparison of the results with that of vbench.
    3. Lack of comparison with other methods and baseline

**Questions:**

See weaknesses.

**Details Of Ethics Concerns:**

This work includes a human annotation process. Although implemented by a certified annotation company, some details are missing.
Providing details about instructions, cases, annotation UI, and the results of quality examination may facilitate readers' understanding and ensure the rationality of the process.

---

### Official Review · Reviewer_dzVd · 2025-10-21

**Soundness:** 2
**Presentation:** 2
**Contribution:** 2
**Rating:** 2
**Confidence:** 4

**Summary:**

The paper introduces GQA (Generation Quality Assessment), a human-aligned benchmark for evaluating AI-generated videos, focusing on three key dimensions: video-text consistency, realism, and traditional quality. It presents a comprehensive dataset of 9,474 videos from 35 generative models, evaluated across 3 main dimensions.  The paper also presents a baseline model that demonstrates the feasibility of using the GQA dataset for training quality assessment models. Despite its strengths in human-centered evaluation and broad model coverage, the paper highlights challenges such as data imbalance and the need for improvements in long-term coherence assessment.

**Strengths:**

1. Human-Centered Evaluation: The introduction of the GQA dataset is a significant strength, as it aligns AI-generated video assessments with human perceptual judgment.
2. Wide Model Coverage: The dataset includes evaluations from 35 generative models, representing a broad spectrum of advancements in the video generation field.
3. Baseline Model Development: A baseline model using VideoMAE and CLIP is presented to show the feasibility of using GQA for supervised learning tasks, demonstrating the utility of the dataset for further research and model improvement.
4.  Comprehensive Evaluation Dimensions: The paper systematically categorizes 32 evaluation dimensions into three core dimensions: Video-Text Consistency, Realism, and Traditional Quality.

**Weaknesses:**

1. Concerns on Annotation Quality: The paper does not explicitly validate the quality of the data annotation using metrics like inner annotator agreement or SRCC (Spearman Rank Correlation Coefficient) between different annotators. While the authors describe a quality control mechanism involving a review panel and random audits of annotations, there is no direct mention of calculating the inter-rater reliability or measuring SRCC to quantify the consistency between annotators.
2. Imbalance in Data Distribution: There is a significant imbalance in the distribution of annotation samples across different dimensions, which could affect model performance and convergence, particularly for rare evaluation criteria like rigid body collisions.
3. Lack of Cross-Dataset Validation: The paper does not include cross-dataset experiments to verify the generalizability of the dataset and model. The results primarily focus on the performance within the GQA dataset, without evaluating how well the models trained on GQA perform on other datasets.
4. Lack of  Ablation Studies: There is a lack of ablation studies that would provide insight into the contribution of different components of the model or evaluation framework.
5. Unverified Model Effectiveness: The effectiveness of the proposed model has not been thoroughly validated. While the paper presents a baseline model, it does not provide detailed performance validation or comparisons with other state-of-the-art methods across different evaluation tasks.
6. Insufficient Information in Tables and Figures: Table 2 and Figure 6 include valuable information, but their content would be better suited for supplementary materials rather than the main body of the paper.

**Questions:**

1. Can you provide details on whether inter-rater reliability (such as SRCC) was calculated to assess the consistency of annotations across different annotators?
2. Why there is a significant imbalance in annotation samples across different dimensions?
3. The results in the paper are primarily based on the GQA dataset. Have you conducted any cross-dataset validation to evaluate the generalizability of the models trained on GQA?
4. Could you provide a more detailed comparison of the performance of the proposed model with existing methods, particularly in terms of SRCC/PLCC?
5. What are the contribution of different components of the model and the impact of different features (e.g., video-text consistency, realism, etc.) on the overall performance?

---

### Official Review · Reviewer_McvQ · 2025-10-27

**Soundness:** 1
**Presentation:** 1
**Contribution:** 1
**Rating:** 0
**Confidence:** 5

**Summary:**

This paper introduces GQA, a new benchmark dataset for the quality assessment of AI-generated videos. The authors define a hierarchical set of 32 evaluation metrics across three main categories: Video-Text Consistency, Realism, and Traditional Quality. The dataset annotation follows a two-stage process: an initial coarse categorization of videos followed by fine-grained pairwise comparisons to derive ranked scores. Additionally, the paper presents a proof-of-concept baseline model, which uses a pre-trained VideoMAE for video features and CLIP for text features to predict quality scores for each of the 32 dimensions. The main contribution of this work is the dataset itself, intended to facilitate more structured and human-aligned evaluation of generative video models.

**Strengths:**

Important Problem: The paper addresses a critical and timely problem: the need for better, more fine-grained, and human-aligned evaluation of AI-generated videos. The limitations of existing metrics are well-recognized, and the goal of creating a multi-dimensional benchmark is laudable.

Comprehensive Taxonomy: The authors have put considerable effort into defining a detailed taxonomy of 32 evaluation dimensions. This hierarchical structure (Consistency, Realism, Quality) is a reasonable way to conceptualize the different facets of video quality.

**Weaknesses:**

Fundamentally Flawed Dataset Construction: Tying evaluation dimensions to specific prompt keywords is a critical design flaw. A video's "Fluid Motion Realism" should be evaluable regardless of whether the prompt contained the word "water" or "river". This approach tests a model's ability to handle specific prompts rather than its general capability to produce realistic videos, making the resulting dataset of limited use for general evaluation.

Lack of Novelty: The paper does not introduce any novel methods. The annotation scheme is a variant of standard pairwise comparison. The baseline model is a simplistic combination of existing, off-the-shelf components. The core ideas are either borrowed from prior work (e.g., multi-dimensional evaluation from VBench) or are not sufficiently developed (e.g., the "prompt-adaptive metric selection").

Insufficient Detail and Unsubstantiated Claims: The paper makes claims in the abstract that are not supported in the main text. For instance, the "prompt-adaptive metric selection mechanism" is a core advertised feature but is never explained, designed, or tested. The process of converting pairwise comparisons into final scores is also not detailed, which is a critical piece of information for a dataset paper.

Poor Baseline and Lack of Comparison: The provided baseline is very weak and is not compared against any other state-of-the-art video evaluation models (e.g., VideoScore, VideoReward, etc.). Without such comparisons, it is impossible to situate the difficulty of the GQA dataset or to understand if the baseline's poor performance is due to the dataset's challenges or the model's simplicity. The paper essentially exists in a vacuum, without proper context within the current literature.

**Questions:**

Could you please clarify the rationale behind tying the data collection for specific evaluation dimensions to the content of the text prompts? This seems to introduce a strong bias. For example, how would you evaluate "Fluid Motion Realism" on a video that realistically portrays a waterfall, but was generated from a prompt that did not explicitly mention water?

The abstract mentions a "prompt-adaptive metric selection mechanism," which sounds like a key component of your framework. However, this mechanism is not described anywhere in the methodology or experiments. Could you please explain what this is and where it is detailed in the paper? If it is not part of the current work, it should be removed from the abstract.

Could you provide a detailed explanation of the algorithm used to convert the pairwise comparison results (A is better than B, B is equivalent to C, etc.) into the final numerical scores? What ranking algorithm (e.g., Bradley-Terry, Elo) was used, and how was the mapping to a continuous or discrete score range performed?

Why did you choose not to compare your baseline model with any existing video quality assessment models on your new dataset? Such a comparison would be crucial to benchmark the difficulty of GQA and to understand its relationship with existing evaluation paradigms.

---

### Official Review · Reviewer_FH31 · 2025-10-31

**Soundness:** 3
**Presentation:** 3
**Contribution:** 3
**Rating:** 4
**Confidence:** 3

**Summary:**

This paper introduces GQA, a new dataset and evaluation framework for assessing the quality of AI-generated videos. The authors argue that existing automated metrics often fail to align with human perceptual judgment. To address this, they propose a human-centric approach. The core contrubution is GQA dataset, featuring videos from various generative models and are evaluated by human annotators across 32 distinct quality dimensions. In addition, a two-stage annotation process that combines coarse-grained classification with fine-grained pairwise comparisons is proposed to improve both the efficiency and accuracy of human labeling.

**Strengths:**

1. Adopting wide evaluation dimensions categorized into Video-Text Consistency, Realism & Plausibility, and Traditional Quality.
2. Propose an innovative and efficient two-stage annotation methodology using coarse-grained classification and fine-grained pairwise comparisons.
3. The newly-created datasets include diverse videos from 35 generative models.
4. The paper is well-written, logically structured, and easy to follow.

**Weaknesses:**

1. Imbalance data: the data ranges from over 7,000 samples for some metrics to as few as 100 for others. It may be very difficult to train reliable, generalizable automated models.
2. It does not report any quantitative metrics for inter-annotator agreement.
3. The videos and prompts are from the VBench dataset, limiting the diversity of content and prompts in GQA.
4. The baseline models are trained separately for each of the 32 dimensions, which are computationally inefficient.

**Questions:**

1. How to solve or mitigate the imbalance data?
2. Does it will be help when expanding the prompt diversity?
3. Why not design a unified model to learn to assess video quality holistically?

---

### Note · Authors · 2025-11-12

I have read and agree with the venue's withdrawal policy on behalf of myself and my co-authors.